# VEGFR2 Blockade Improves Renal Damage in an Experimental Model of Type 2 Diabetic Nephropathy

**DOI:** 10.3390/jcm9020302

**Published:** 2020-01-21

**Authors:** Carolina Lavoz, Raul R. Rodrigues-Diez, Anita Plaza, Daniel Carpio, Jesús Egido, Marta Ruiz-Ortega, Sergio Mezzano

**Affiliations:** 1Laboratorio de Nefrología, Facultad de Medicina, Universidad Austral de Chile, Bueras 1003, Valdivia, Chile; carolavoz@gmail.com (C.L.); anitaplazaflores@yahoo.com (A.P.); dcarpiop@gmail.com (D.C.); mezzano.sergioa@gmail.com (S.M.); 2Cellular and Molecular Biology in Renal and Vascular Pathology Laboratory, Fundación Instituto de Investigación Sanitaria-Fundación Jiménez Díaz-Universidad Autónoma Madrid, Reyes Católicos 2, 28040 Madrid, Spain; rrodriguez@fjd.es; 3Red de Investigación Renal (REDINREN), 28040 Madrid, Spain; 4Unidad Microscopía Electrónica, Vice-Rectoria de Investigación, Desarrollo y Creación Artística, Universidad Austral de Chile, Independencia 631, Valdivia, Chile; 5Renal, Vascular and Diabetes Research Laboratory. Fundación Instituto de Investigación Sanitaria-Fundación Jiménez Díaz-Universidad Autónoma Madrid, 28040 Madrid, Spain; jegido@fjd.es; 6Spanish Biomedical Research Centre in Diabetes and Associated Metabolic Disorders (CIBERDEM), 28040 Madrid, Spain

**Keywords:** VEGFR2, VEGFA, GREMLIN, inflammation, podocytes, diabetes, diabetic nephropathy, tubular cells

## Abstract

The absence of optimal treatments for Diabetic Nephropathy (DN) highlights the importance of the search for novel therapeutic targets. The vascular endothelial growth factor receptor 2 (VEGFR2) pathway is activated in experimental and human DN, but the effects of its blockade in experimental models of DN is still controversial. Here, we test the effects of a therapeutic anti-VEGFR2 treatment, using a VEGFR2 kinase inhibitor, on the progression of renal damage in the BTBR ob/ob (leptin deficiency mutation) mice. This experimental diabetic model develops histological characteristics mimicking the key features of advanced human DN. A VEGFR2 pathway-activation blockade using the VEGFR2 kinase inhibitor SU5416, starting after kidney disease development, improves renal function, glomerular damage (mesangial matrix expansion and basement membrane thickening), tubulointerstitial inflammation and tubular atrophy, compared to untreated diabetic mice. The downstream mechanisms involved in these beneficial effects of VEGFR2 blockade include gene expression restoration of podocyte markers and downregulation of renal injury biomarkers and pro-inflammatory mediators. Several ligands can activate VEGFR2, including the canonical ligands VEGFs and GREMLIN. Activation of a GREMLIN/VEGFR2 pathway, but not other ligands, is correlated with renal damage progression in BTBR ob/ob diabetic mice. RNA sequencing analysis of GREMLIN-regulated genes confirm the modulation of proinflammatory genes and related-molecular pathways. Overall, these data show that a GREMLIN/VEGFR2 pathway activation is involved in diabetic kidney disease and could potentially be a novel therapeutic target in this clinical condition.

## 1. Introduction

Diabetic Nephropathy (DN) is a major microvascular complication of diabetes. The epidemic increase of type 2 diabetes is the main cause of Chronic Kidney Disease (CKD) worldwide, leading to end-stage renal disease (ESRD) and premature death. The management of DN focuses on the strict control of hyperglycemia and hypertension as well as the adequate inhibition of the renin-angiotensin-aldosterone system [1]. Recently, two novel antidiabetic drugs, such as the SGLT-2 (sodium-glucose cotransporter-2) inhibitors and GLP1 (Glucagon-like peptide-1) agonists, have shown marked beneficial effects on cardiovascular and renal events, and decreased mortality [2]. However, better therapeutic options are still needed. Among potential novel therapies, the anti-inflammatory targets have been proposed as promising options [3].

The vascular endothelial growth factor receptor 2 (VEGFR2) is expressed in several renal cells, including podocytes and tubular epithelial cells [4,5]. This receptor is activated in experimental and human renal diseases [6], including DN [7]. Most of the studies about VEGFR2 signaling have been done in cell carcinomas [8]. The canonical VEGFR2 ligand is VEGF-A, the most potent angiogenic factor [9]. VEGF-A binds to VEGFR2, and the co-receptors neuropilin 1 and 2 (NRP1 and NRP2), while VEGFR1 functions mostly as a decoy [10]. In physiological conditions, VEGFs regulate endothelial function and the glomerular filtration barrier in the kidney. However, deregulation of VEGF/VEGFR signaling occurs in pathological conditions, including DN [9]. Cultured podocytes exposed to high glucose can produce VEGF-A in a nephrin-dependent regulated manner [11,12]. Regarding human DN, upregulation of glomerular VEGF-A has been associated to glomerular basement membrane expansion and capillary wall deformation [13]. Preclinical studies manipulating VEGF-VEGFR signaling in DN had reported contradictory results, showing beneficial or deleterious effects depending on the study [9].

Moreover, the complexity of this system has recently been increased by the discovery of a novel VEGFR2 ligand; GREMLIN [7,14]. This protein is a member of the TGF-β superfamily and acts as a Bone Morphogenetic Protein (BMP) antagonist mainly in embryonic development [15]. *GREM-1* is a developmental gene, expressed during development, silenced in the adult and re-expressed under pathological conditions, including renal diseases [15]. Several groups, including ours, have proposed GREMLIN as a potential therapeutic target in DN [15]. In the earliest studies done by Brady et al. [16] *grem-1* was identified as an upregulated gene in rat diabetic kidneys and in cultured mesangial cells stimulated with high glucose levels. In renal biopsies of DN patients, elevated GREMLIN expression, at gene and protein levels, was found, predominantly in tubular epithelial cells [17]. Moreover, increased urinary GREMLIN levels were described in patients with type 1 diabetes and DN compared to controls [18]. Several preclinical studies in the mice model of streptozotocin (STZ)-induced type 1 diabetes also support the involvement of GREMLIN in DN. Accordingly, studies in genetic modified mice showed that allelic depletion of *grem1* elicited renal protection in response to STZ [19]. Moreover, *grem-1* gene silencing diminished proteinuria, renal fibrosis and cell proliferation/apoptosis in this model [20]. To contrast, tubular overexpression of *GREM-1* in transgenic mice aggravated renal damage induced by STZ [21]. Despite the fact of these outcomes, future studies using alternative GREMLIN blockade strategies are still necessary to translate these promising results into clinical studies. In vivo studies have demonstrated that VEGFR2 is a functional receptor for GREMLIN in the kidney [7,15]. Importantly, in renal biopsies of DN patients co-localization of GREMLIN and activated VEGFR2 has been described [7], supporting the pathogenic role of GREMLIN/VEGFR2 axis in human DN.

An important obstacle in the study of DN and its complications is the lack of robust animal models that replicate the key features of human diabetes to test novel therapeutic tools [22]. Actually, the above commented discrepancies about VEGF-VEGFRs could be due to the differences between preclinical mice models (mainly STZ-induced diabetes) and human DN [9]. One of the most recommended experimental models to study DN is the BTBR ob/ob (Black and Tan Brachyury Obese, characterized by a leptin deficiency mutation) mice [23]. Therefore, the aim of this work is to investigate the effect of the VEGFR2 signaling blockade in the progression of DN in the BTBR ob/ob model, evaluating the contribution of canonical (VEGFs) and non-canonical ligands (GREMLIN) to VEGFR2 pathway activation. To this aim VEGFR2 activation is blocked using a VEGFR2 kinase inhibitor as a therapeutic approach, starting the treatment after kidney disease development.

## 2. Experimental Section

### 2.1. Design of the Experimental Model of Diabetic Nephropathy and Characterization

The general principles of laboratory-animal care were followed, and the mice were euthanized after anesthesia administration in accordance with the protocols approved by the Ethics Committee for Animal Experiments of the Universidad Austral de Chile (Permit No. 245-2016) and according to National Institutes of Health guidelines. The establishment and care of the BTBR ob/ob diabetic and obese mice colony (referred to here as “diabetic mice”) and their corresponding controls (BTBR wild type (WT) littermate mice) have been previously described [24]. These mice rapidly develop morphologic renal lesions characteristic of both early and advanced human DN [25].

Male BTBR ob/ob diabetic mice and their corresponding controls were euthanized serially every 2 weeks, starting at week 4 and up until 20 weeks of age (*n* = 6 for each group).

Body weight was checked weekly. Blood glucose and serum creatinine levels were measured by the Accu-check^®^ Performa (Roche, Mannheim, Germany) and Jaffe reaction (Creatinine liquicolor, HUMAN diagnostics, Wiesbaden, Germany), respectively, executing a caudal vein puncture every week. Spot urine samples were collected once a week from all mice and analyzed for albumin by ELISA (Enzyme-Linked ImmunoSorbent Assay) (ALPCO Immunoassays, Salem, NH, USA) and for creatinine by Jaffe reaction (Creatinine liquicolor, HUMAN diagnostics) to obtain the urine albumin/creatinine ratio.

Occurring at the time of euthanasia, serum was collected, and mice were anesthetized with 2% 2,2,2-tribromethanol (Sigma, Darmstadt, Germany) dissolved in 2-methyl-buthanol (Sigma). The kidneys were removed, decapsulated, and cut along the sagittal plane. A portion of the left kidney was fixed in 4% formaldehyde or 2% glutaraldehyde while the right kidney was immediately frozen in liquid nitrogen and processed for RNA and protein extraction. 

### 2.2. VEGFR2 Pharmacological Treatment

The VEGFR2 signaling pathway was blocked with the VEGFR2 kinase inhibitor SU5416. The VEGFR blocking treatment was started in the fifteenth week of life when the BTBR ob/ob mice have already developed the characteristic lesions of DN (therapeutic approach). Straight away, mice were randomly distributed in 2 groups (SU5416-treated and vehicle-treated; *n* = 6–8 mice per group). SU5416 was administered intraperitoneally (i.p.) to mice at a dose of 0.1 mg per mouse (dissolved in vehicle, DMSO less than 0.01%), 3 times a week, for 5 weeks, and then euthanized at 20 weeks. During some experiments, these mice were compared with untreated diabetic mice or non-diabetic (BTBR WT littermate) mice of the same age. The designed experiment had 3 groups: (1) non-diabetic (controls; *n* = 8 animals), (2) vehicle-treated diabetic (untreated-diabetic; *n* = 6 animals), and (3) SU5416-treated diabetic mice (treated diabetic; *n* = 6 animals).

### 2.3. Mice Model of Gremlin-Induced Renal Damage

The model of intra-renal parenchymal injection of Gremlin was done in 3-month-old female C57BL/6 mice at the IIS-Fundación Jimenez Diaz. All animal procedures were performed according to the guidelines of animal research in the European Community under prior approval by the Animal Ethics Committee of the Health Research Institute IIS-Fundación Jiménez Díaz. The model was performed under isoflurane-induced anesthesia. Mice received an injection of 50 ng Gremlin murine recombinant protein in the left kidney as described [7]. A sham-operated group with saline injection also was employed (*n* = 6–8 animals per group). Renal tissue was obtained and stored as described above.

### 2.4. Protein Studies

Total proteins from frozen renal tissues were isolated in T-PER Reagent (Thermo Scientific, Rockford, IL, USA). Protein levels were quantified using a PierceTM BCA protein assay kit (Thermo Scientific). Renal protein levels were evaluated by western blot. Proteins (100 µg/lane) were separated on 10–12% polyacrylamide-SDS gels under reducing conditions. Protein quality and efficacy of transfer were evaluated by Ponceau red staining (not shown). Primary antibodies were detected with an HRP-conjugated secondary antibody, developed by LuminataTM Forte (Millipore, Billerica, MA, USA) and scanned using the G:Box Chemi XRQ (Syngene, Frederick, MD, USA). GAPDH (Glyceraldehyde 3-phosphate dehydrogenase) was used as a loading control. The following primary antibodies were employed [dilution]: phosphorylated-VEGFR2 (Tyr 996) (Santa Cruz, 16629) [1:500] and GAPDH (Chemicon Int, MAB374 [1/5000]).

### 2.5. Histological Analysis and Immunohistochemistry

The samples fixed in 4% formaldehyde were embedded in paraffin and cut into 4-μm tissue sections for further histological (PAS/Masson) and immunohistochemistry (IHC) studies. The tissue damage score was calculated as described [26], including the degree of glomerular sclerosis, increased mesangial matrix, hyalinosis, tubular casts, acute tubular damage, and tubular atrophy, as well as the presence of interstitial inflammatory cells. The tissue fixed in 2% glutaraldehyde (Merck, Darmstadt, Germany) was post-fixed with 1% osmium tetroxide (Ted Pella Inc., Hedding, CA, USA) included in resin EMBed-812 (EMS, Hatfield, PA, USA), cut, stained and observed under an AURIGA Compact Scanning Transmission Electron Microscope (Zeiss, Oberkochen, Germany). Glomerular and tubulo-interstitial lesions were graded according to their histopathological score (from 0 to 4) as previously described [26].

IHC for detection of Wilms’ tumor protein 1 (WT-1) as a podocyte marker was performed following a heat-induced antigen retrieval system (Tris-base 10 mM, EDTA 1 mM, 0.05% Tween 20, pH 9.0) for 10 min in a microwave oven. Sections were incubated overnight with Monoclonal Mouse Anti WT-1 protein Clone 6F-H2, M3561 (dilution: 1:100, DAKO, Carpinteria, CA, USA) followed by incubation with the M.O.M. Immunodetection kit (PK 2200, Vector, Burlingame, CA, USA) and ImmPACT DAB Peroxidase Substrate (Vector). 

Interstitial infiltrating cells were detected by means of F4/80 (monocytes/macrophages) and CD3 (T lymphocytes) antibodies, as described [27]. F4/80 was detected using the MA1-91124 antibody (dilution: 1/100, Thermo) followed by Immpress Reagent Kit (MP 7444, Vector), and CD3 (A 0452 antibody, dilution: 1:200, DAKO) were detected using Trilogy epitope retrieval (Cell Marque, Rocklin, CA, USA) and followed by horseradish peroxidase streptavidin (dilution 1:500, SA-5004 Vector), revealed with DAB SK4105 (Vector), and finally counterstained with hematoxylin, as previously described [27]. Next, endogenous peroxidase was blocked and sections were incubated for 30 min at room temperature or o/n at 4 °C with primary antibodies. Following this, slides were treated with the EnVision™ DuoFLEX Doublestain System using 3,3’-diaminobenzidine as a chromogen. Sections were counterstained with Carazzi’s hematoxylin and evaluated by optical microscopy. 

Image analysis and quantification of the IHC signals were performed using the KS300 imaging system, version 3.0 (Zeiss). Regarding each sample, the mean staining area was obtained by analysis of twenty randomly chosen fields (×200 magnification) using Image-Pro Plus software. Data were expressed as positive-stained area compared to total area analyzed.

Podocyte involvement was calculated by enumerating podocyte nuclei stained for WT-1 positive glomerular cells in 25 glomeruli. 

### 2.6. Gene Expression Studies

Total RNA from renal tissue was isolated with TRIzol reagent (AmbionTM, Carlsbad, CA, USA) according to the method provided by the manufacturer and treated with DNase I to remove potential contamination with genomic DNA. cDNA was synthesized by the ImProm-II TM Reverse Transcription System (Promega, Madison, WI, USA) using 2 µg of total RNA primed with random hexamer primers. Quantitative gene expression analysis was performed on a Rotor-Gene Q (Qiagen, Hilden, Germany) using primers designed by IDT^®^ (Coralville, IA, USA) and the reagent KAPA SYBR^®^ FAST Universal 2X qPCR Kit Master Mix (Kapa Biosystems, Wilmington, MA, USA) to determine the expression levels of candidate genes (Table 1). Primers used for qPCR are shown in Table 1. PCR product specificity was verified by melting curve analysis, and all real-time PCR reactions were performed in triplicate. The 2-ΔΔCT method was used to analyze the relative changes in gene expression levels.

### 2.7. RNA Sequencing Studies

Libraries were prepared according to the instructions of the Kit “NEBNext Ultra Directional RNA Library Prep kit for Illumina” (New England Biolabs, United Kingdom), following the protocol “Poly (A) mRNA Magnetic Isolation Module”. RNA quality was evaluated according to RNA integrity numbers (RIN, 8.5-9), quantified using a RNA 6000 nano LabChipin an Agilent 2100 Bioanalyzer (ranges 6–7. Agilent, CA, United States). The input yield of total RNA to start the protocol was 900 ng. The fragmentation time used was 8–15 min according to RIN values. The rest of the protocol was performed according to manufacturer instructions.

The obtained libraries were validated and quantified by DNA7500 LabChip kit (Agilent 2100 Bioanalyzer). An equimolecular pool (5 mice per group) of libraries was titrated by quantitative PCR using the “Kapa-SYBR FAST qPCR kit forLightCycler480” (Kapa Biosystems, MA, USA) and a reference standard for quantification. The pool of libraries was denatured prior to being seeded on a flow cell at a density of 2.2 pM, where clusters were formed and sequenced using a “NextSeq™ 500 High Output Kit” (Illumina, CA, USA), in a 1 × 75 single read sequencing run on a NextSeq500 sequencer (Illumina). An amount of approximately 50 million pass-filter reads (range 42–54) were produced for each pool of samples, which were used for further bioinformatics analysis.

FASTq files were analyzed using the XploreRNA service from Exiqon–Qiagen (Qiagen GmbH, Hilden, Germany). The obtained results were studied using the Functional Annotation Tool of David database [28,29]. Although our samples were analyzed as pools, limiting the statistical power of our findings about differentially expressed genes [28], we used them as a hypothesis based design.

## 3. Results

### 3.1. VEGFR2 Kinase Inhibition Improves Renal Function in Experimental Diabetic Nephropathy

The evaluation of phosphorylated VEGFR2 renal levels showed activation of this pathway in diabetic kidneys compared to their corresponding controls (BTBR WT mice of the same age) (Figure 1). Next, we confirmed that the beneficial effects of SU5416 treatment were due to the blockade of VEGFR2 pathway activation. Seen in SU5416-treated diabetic mice was a significant diminution in phosphorylated VEGFR2 levels compared to untreated BTBR ob/ob mice (Figure 1).

Then, the effect of the VEGFR2 blockade on renal injury on experimental diabetes in BTBR ob/ob mice was investigated. To this aim, the albumin/creatinine ratio (ACR) was measured in spot urine samples. VEGFR2 treatment improved albuminuria, as shown by a significant decrease in ACR values during all periods of study compared to the untreated diabetic group. Occurring at 20 weeks, there was a 21% reduction versus untreated diabetic mice (643.80 + 12.26 ug/mg versus 816.09 + 32.57 ug/mg; *p* < 0.05 versus untreated diabetic mice) (Figure 2a). These data suggest that the VEGFR2 kinase inhibitor ameliorates the albuminuria in experimental type 2 diabetes associated with obesity.

Regarding diabetic mice, blood glucose levels were significantly elevated at 6 weeks, and increased to higher blood glucose levels at the end of the study (20 weeks), compared to control non-diabetic mice (593.5 + 2.1 mg/dL versus 168.3 ± 3.2 mg/dL; *p* < 0.05 versus control non-diabetic mice). There were no changes on blood glucose levels throughout the study compared to untreated control diabetic mice (598 + 1.0 versus 168.3 + 3.2; *p* < 0.05 versus control non-diabetic mice) (Figure 2b) in response to VEGFR2 treatment.

BTBR ob/ob mice presented a significant increase in body weight compared to WT (70.32 ± 3.89 versus 37.0 ± 1.4 versus control non-diabetic mice) that was unchanged in mice treated with the VEGFR2 inhibitor (Figure 2c).

### 3.2. VEGFR2 Kinase Inhibition Ameliorates Renal Damage in Experimental Diabetic Nephropathy

The effect of the VEGFR2 blockade on renal structural changes was studied at the end of the study. Periodic acid–Schiff staining revealed an increase in mesangial matrix, hyalinosis, and interstitial cellular infiltration in BTBR ob/ob diabetic mice compared with control (BTBR WT) mice. The evaluation of morphological lesions by PAS staining showed that mice treated with the VEGFR2 inhibitor presented lower renal lesions than untreated diabetic mice (Figure 3a–c). To further assess glomerular damage, electron microscopy analysis was conducted. Diabetic kidneys showed thickening of the glomerular basement membrane, irregular laminations, and focal protrusions in some segments, along with a greater effacement of foot processes, thus resembling lesions found in diabetic patients; these changes were not observed in the control non-diabetic mice (Figure 3d–f,h,i). Diabetic mice presented less glomerular basement membrane thickening (Figure 3d,g) and significantly less mesangial matrix expansion (Figure 3i,j) in response to SU5416-treatment.

Previous studies have found that renal expression of the two biomarkers of renal injury, such as the kidney injury molecule 1 (KIM-1) and neutrophil gelatinase–associated lipocalin (NGAL) were increased in BTBR ob/ob mice [27]. Treatment with the VEGFR2 kinase inhibitor significantly diminished *kim-1* and *ngal* renal gene expression compared to untreated diabetic mice (Figure 4a,b). All these data suggest that a VEGFR2 blockade ameliorates renal damage in experimental diabetes.

### 3.3. VEGFR2 Kinase Inhibition Diminishes Podocyte Damage in Experimental Diabetic Nephropathy

Podocyte damage is a key feature of DN. The loss of podocytes observed in BTBR ob/ob mice was diminished in response to the VEGFR2 blockade, as manifested by a restoration of gene expression levels of *wt-1*, *nphs-1* and *nphs2* (Figure 5a–c), that codify key podocytes proteins. Moreover, VEGFR2 inhibition also increased the number of WT-1 positive cells (Figure 5b,c), showing a protective effect on podocytes.

### 3.4. VEGFR2 Kinase Inhibition Decreases Renal Inflammation in Experimental Diabetic Nephropathy

Next, we further evaluated whether a VEGFR2 blockade could modulate the inflammatory-related process in diabetic kidneys (Figure 6 and Figure 7). Using immunohistochemistry, a decrease in renal inflammatory infiltrating cells (CD3, and F4/80 positive cells) was found in SU5416-treated mice compared to the untreated-diabetic group (Figure 6a,b). Moreover, the gene over-expression of the pro-inflammatory factors *mcp-1*, *rantes*, *il-6* and *il-17a* observed in BTBR ob/ob mice kidneys were downregulated in response to VEGFR2 inhibitory treatment (Figure 7a–d). 

### 3.5. Evaluation of the Renal VEGFR2 Ligands Expression in Experimental Diabetic Nephropathy

Previous studies have described an upregulation of several VEGFR2 ligands in experimental diabetes [9], but there is no available data in the BTBR ob/ob model. Therefore, we evaluated renal gene expression levels of the canonical VEGFR2 ligands over the time-course of the diabetic disease. The renal gene expression of *vegf-a* was significantly increased at 8 weeks, coincidentally with the onset of the diabetes, but was downregulated thereafter, showing no difference to non-diabetic mice (Figure 8a). Renal gene expression of *vegf-d* was only increased at 4 weeks in diabetic mice compared to controls (Figure 8b).

GREMLIN has been proposed as a non-canonical VEGFR2 ligand [7,14]. The evaluation of *grem-1* gene expression levels showed no difference between BTBR ob/ob and BTBR WT mice at early time points and only after 10 weeks was the *grem-1* gene increased in diabetic mice and remained significantly elevated until the end of the study at 20 weeks (Figure 8c).

### 3.6. Gene Expression Levels Regulated by Gremlin in the Kidney, and Their Related Gene Ontology (GO) and Kyoto Encyclopedia of Genes and Genomes (KEGG) Pathway Enrichment Analysis

An RNA-sequencing approach was used to assess renal gene expression changes and provide insight into the molecular mechanisms involved in Gremlin responses in the kidney. To this aim a mice model of renal administration of murine Gremlin recombinant protein was used, as previously described and characterized [7]. A heat map two-way hierarchical clustering of genes and samples showed that Gremlin regulates a large number of genes (Figure 9). Table 2 shows the most deregulated genes that were used to perform the Gene Ontology (GO) enrichment analysis using the DAVID database tools (Table 3) [29,30]. The biological process (BP) analysis indicated mechanical stimuli, wound healing or immune system as the most relevant phenomena. Cellular Component (CC) predictions reflected the potential involvement of genes related to the extracellular matrix region and cell surface among others, while the molecular function (MF) results showed genes related to the extracellular matrix structural constituent or protease binding. Finally, the KEGG pathway enrichment analysis presented several pathways including complement and coagulation cascades, extracellular matrix (ECM)-receptor interaction or the phosphatidylinositol-3-kinase (PI3K)/protein kinase B (AKT) signaling pathway. Although the samples were analyzed as pools, limiting the statistical power of the findings to detect differentially expressed genes [28], the results were used as hypothesis generating experiments. Some of these genes had been previously validated by PCR in [7], as *havcr1* (*hepatitis A virus cellular receptor 1*, also known as *kim-1*) and *lcn2 (lipocalin 2 or Ngal)*, confirming the data of Table 2.

## 4. Discussion

Several previous studies, both in cell cultures and in diabetic animals, have suggested a role for VEGF/VEGFR2 in the development of DN. However, activation or inhibition studies of this pathway have elicited contradictory data, mainly attributed to the experimental diabetic model, the therapeutic modulation (genetic versus pharmacological) and the time-point of therapeutic intervention. Additionally, the biological effects of VEGF/VEGFR2 signaling depend on the amount of VEGF, the VEGF type, and the target cell, among other factors [31]. Here, we describe the renal beneficial effects of the VEGFR2 kinase inhibitor SU5416 in the BTBR ob/ob mice model that recapitulates the renal lesions of human DN [23,24,25]. The administration of SU5416, starting when renal damage was already established, markedly impaired albuminuria and the diabetic structural abnormalities, including mesangial matrix accumulation, GBM thickening and inflammatory cell infiltrates. Accordingly, the downregulation of genes linked to renal damage such as *Kim-1* and *Ngal* further support the protective effects of the VEGFR2 blockade in an experimental type 2 diabetes model. All these data suggest that the pharmacological inhibition of this pathway could be an interesting option to be further explored in the treatment of DN.

Some of the current treatments employed for chronic kidney diseases, including renin angiotensin aldosterone system blockers, can reduce albuminuria and seem to have protective effects on podocytes [32,33,34,35]. Podocytes are very specialized and differentiated cells that can be damaged under high glucose or stress conditions [36], as we observed here in the BTBR ob/ob mice model by a decrease in the number of WT-1+ podocytes in the diabetic kidneys. Disruption of slit diaphragm proteins has been involved in the appearance of albuminuria in DN [36]. Nephrin is one of the most relevant of those proteins, which functions both as a structural component and a signaling protein via PI3K pathway modulation [37]. Several factors involved in renal diabetic damage, such as Angiotensin II, glycated albumin and VEGF, induce nephrin downregulation in cultured podocytes [38,39]. Regarding BTBR ob/ob mice, the treatment with a VEGFR2 kinase inhibitor normalized the changes in gene expression of *Nphs-1* and *Nphs2* and prevented the loss of WT-1+ podocytes. These data suggest that VEGFR2 inhibitors could be added to the array of drugs exerting podocyte protective properties. After intensive or prolonged injury, lost podocytes are replaced by collagens, therefore contributing to glomerulosclerosis and renal function decline [36,40]; therefore, targeting injured podocytes constitutes an attractive therapeutic option. Interestingly, several therapeutic strategies may promote the expression of adaptive endogenous protective factors, such as heat shock proteins that help podocytes to withstand stressors such as high glucose or Angiotensin II elevated levels [41]. However, the role of protective factors in VEGFR2 actions will require future research. Our data clearly show that VEGFR2 kinase inhibition, by preventing podocyte loss through targeting nephrin, could contribute to the reduction of albuminuria and, therefore, exert protective effects in DN.

A VEGFR2 pathway can be activated by several ligands. During the early stages of DN, upregulation of VEGF-A, the canonical VEGFR2 ligand, has been described in different experimental models [9]. Podocyte-specific VEGF164 overexpression in adult mice demonstrated that VEGF-A, activating VEGFR2 in podocytes, can induce reversible changes on the slit diaphragm proteins contributing to the abnormalities in the glomerular filtration barrier, as noted in the diabetic kidney [39]. To contrast, VEGF-A also has shown protective effects in acute models of renal injury, although the mechanisms involved have not been completely elucidated yet [6]. Concerning the BTBR ob/ob model, renal *vegf-a* gene expression was significantly increased at 8 weeks, coincidentally with the onset of the diabetes, but its gene expression was downregulated thereafter, showing no differences to non-diabetic mice. Additionally, *vegf-d* mRNA levels were only elevated at early stages of the DN. These data suggest that VEGF-A could participate in the early stages of diabetes but it is not involved in the progression of renal damage, at least in this model of DN. Accordingly, in STZ-induced DN, the podocyte-specific *vegf-a* deletion increased proteinuria by promoting endothelial injury, accelerating the progression of glomerular injury [42], indicating that the increased VEGF-A observed during the early stages of DN may be a compensatory mechanism that could protect renal function. However, additional studies testing VEGF-A modulation in the initial phases of BTBR ob/ob should be addressed to validate this hypothesis.

Another VEGFR2 ligand is GREMLIN [7,14]. Regarding the BTBR ob/ob mice model, there is an upregulation of *grem-1* at 10 weeks, remaining elevated thereafter. Importantly, at 10 weeks BTBR ob/ob mice also presented upregulation of the renal damage biomarkers *Kim-1* and *N-gal* [27]. Our in vivo data showing coincident *grem-1* overexpression with the development of renal damage, support the idea of GREMLIN as a key VEGFR2/ligand mediator of renal damage progression in diabetes [19,20]. Several in vitro studies have previously suggested a role of GREMLIN in diabetes. The incubation in high glucose cultured conditions have shown to increase *grem-1* gene expression and protein production in podocytes [43], mesangial [16] and tubuloepithelial cells (unpublished observations). Moreover, local production of GREMLIN can act on resident and infiltrating cells contributing to renal damage progression [15]. Essentially, in vitro studies in cultured podocytes have shown that GREMLIN elicited phenotype changes, including a decrease and rearrangement of nephrin and synaptopodin expression [43,44]. Concerning mesangial cells, GREMLIN increased cell proliferation and ECM production via ERK1/2 pathway regulation [45]. However, the receptor involved was not investigated in those studies.

Although classically DN has been considered as a glomerular disease, emerging evidence of tubulointerstitial injury has been described in experimental diabetes, even preceding glomerular dysfunction [46]. Additionally, high-glucose-stressed podocytes can promote pathological albuminuria which, in turn, causes proximal tubular cell injury and activates proximal tubular cells to secrete proinflammatory and profibrotic factors, supporting the connection of glomerular and tubular damage in DN [47]. Basically, we have described that GREM-1-specific overexpression in tubular cells accelerates STZ-induced experimental DN, causing podocytopenia and decreased expression of podocin [45]. Interestingly, in human DN biopsies we have found activation of VEGFR2 and overexpression of GREMLIN in tubular epithelial cells [7], supporting the idea that in pathological situations the elevated production of GREMLIN could activate VEGFR2 in tubular cells to promote renal damage. Regarding cultured tubular epithelial cells, GREMLIN caused phenotype changes, including the loss of epithelial properties and the acquisition of mesenchymal markers, producing an aberrant secretome that includes pro-inflammatory and profibrotic proteins [48,49], the main characteristics of partial epithelial to mesenchymal transition (EMT) [47]. This partial EMT has been proposed as the process by which GREMLIN could contribute to renal damage progression. Fundamentally, GREMLIN actions on tubular epithelial cells are mediated by VEGFR2 signaling pathway activation, including EMT and regulation of mediators of injury, inflammation and fibrosis [7,50,51]. Therefore, a blockade of VEGFR2 signaling in the kidney could exert anti-inflammatory and anti-fibrotic effects by targeting GREMLIN actions in tubular cells, as described in other experimental models of renal injury, such as unilateral ureteral obstruction [51], and extended here to experimental diabetic nephropathy.

Taking into account the tremendous importance of inflammation in the genesis and progression of DN, several strategies have been implemented during the last years to tackle this issue [3]. Many evidences suggest that GREMLIN also can be considered as a proinflammatory factor. GREMLIN regulates renal inflammation by the activation of the nuclear factor κB (NF-κB) pathway and upregulation of proinflammatory mediators via the VEGFR2 signaling system [7]. Different NF-κB inhibitors have exerted beneficial effects on experimental diabetes [3], but better targets are needed. Another important pathway involved in DN and activated by the GREMLIN/VEGFR2 signaling axis is NOTCH [50]. Increased expression and activity of NOTCH genes have been described in podocytes of DN in animals and patients [52]. Regarding human DN, co-expression of GREMLIN, and activated NOTCH components have been found, mainly in tubuloepithelial cells [53]. Specific podocyte genetic modulation or pharmacological inhibition have demonstrated the role of this pathway in DN [54]. Treating diabetic rats with a γ-secretase inhibitor, that blocks NOTCH activation, decreased albuminuria, normalized *vegf* and *nephrin* expression, and ameliorated kidney disease [55]. Additionally, GREMLIN also can be produced by infiltrating macrophages, as found in glomerular crescents of human kidney biopsies of patients with ANCA-associated crescentic glomerulonephritis [56]; therefore, strategies preventing macrophage infiltration in DN [3] also can exert beneficial effects by targeting GREMLIN actions.

The molecular mechanisms involved in GREMLIN-induced renal damage are not completely elucidated. Using an RNA-sequencing approach we have evaluated gene expression changes induced by GREMLIN in the kidney. Among the most upregulated genes, *Kim-1* and *Ngal*, are present, confirming the previously described role of Gremlin in renal damage and inflammation [7]. Importantly, proinflammatory genes and ECM-related genes also were elevated. Moreover, the GO enrichment analysis of these data emphasizes the role of wound healing as one of the biological processes implicated in Gremlin-mediated responses, which could be related to GREMLIN-induced phenotype transformations previously described in different cell types [7,57,58,59,60,61]. Moreover, the ECM region and structural constituent are the main highlighted terms of cellular components and molecular function, respectively. In vitro studies suggested that GREMLIN can participate in fibrotic diseases by modulating the TGF-β/Smad pathway [49], or by acting as a BMP antagonist [60,62]. All these data support the role of GREMLIN in the regulation of inflammatory and fibrotic processes. Finally, the KEGG pathway enrichment analysis further emphasizes the PI3K-Akt signaling pathway in GREMLIN signaling. Downstream activation of VEGFR2 includes ERK1/2, p38, Src and Akt pathways [8]. Seen in retinal pigment epithelial cells, GREMLIN, via VEGFR2, activates Akt-mTORC2 (mammalian target of rapamycin complex 2) signaling, and promotes cell proliferation, migration and VEGF production [63]. Commented above, PI3K is a key pathway in Nephrin signaling [37]. The PI3K-AKT-mTOR pathway leads to intracellular metabolic changes, involved in metabolic reprogramming such as upregulation of glycolytic enzymes, and disruption of mitochondrial function. Studies targeting PI3K-AKT-mTOR in malignancies have afforded promising results [64,65,66,67], and could be relevant to DN [68]. Accordingly, nephroprotective effects of the old anti-diabetic drug metformin, mediated via the AMPK/mTOR signaling axis, have been recently described, including diminution of apoptosis [69].

Overall, we demonstrate that the activation of the GREMLIN/VEGFR2 pathway plays an important role in the pathogenesis of diabetic nephropathy. The marked reduction of albuminuria, renal lesions and inflammation-related pathways in a type 2 diabetic mice model by the therapeutic pharmacological inhibition of the VEGFR2 pathway, using a VEGFR2 kinase inhibitor, suggests that this GREMLIN/VEGFR2 pathway could be a feasible therapeutic target for DN. Future development of more specific targets of this pathway should be investigated.

## Figures and Tables

**Figure 1 jcm-09-00302-f001:**
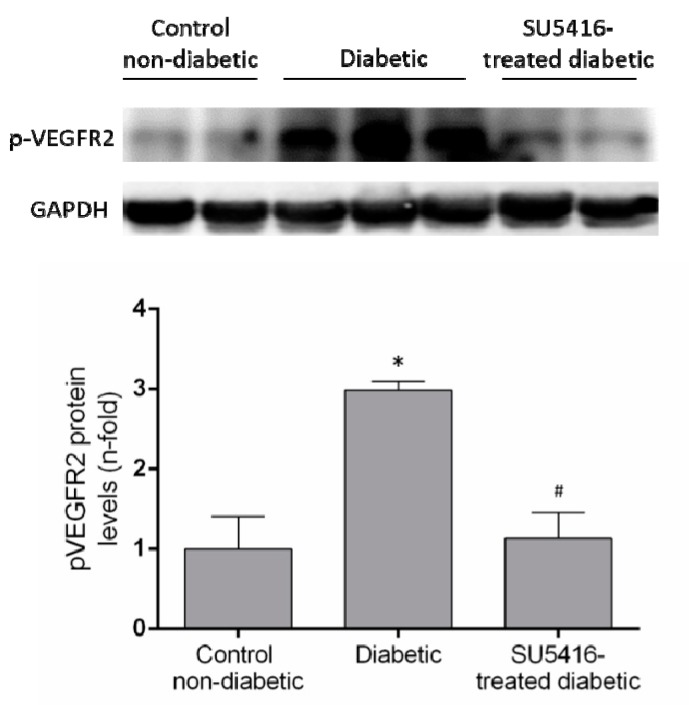
VEGFR2 activation in the BTBR ob/ob animal model. The BTBR ob/ob mice were treated with the VEGFR2 kinase inhibitor SU5416 (i.p, 0.1 mg/mouse, 3 times a week) (SU5416-treated diabetic group) or vehicle (diabetic group), starting at 15 weeks and followed until 20 weeks. BTBR WT mice of the same age were used as a non-diabetic control group. To evaluate VEGFR2 pathway activation, VEGFR2 phosphorylated renal expression levels (*p*-VEGFR2) were determined by western blot. Data of *p*-VEGFR2 levels were obtained from densitometric analysis, as ratios versus corresponding GAPDH, expressed as n-fold over control (considered 1). Figures show a representative western blot and data as mean ± SEM of 6–8 mice per group. * *p* < 0.05 versus control non-diabetic, # *p* < 0.05 versus diabetic.

**Figure 2 jcm-09-00302-f002:**
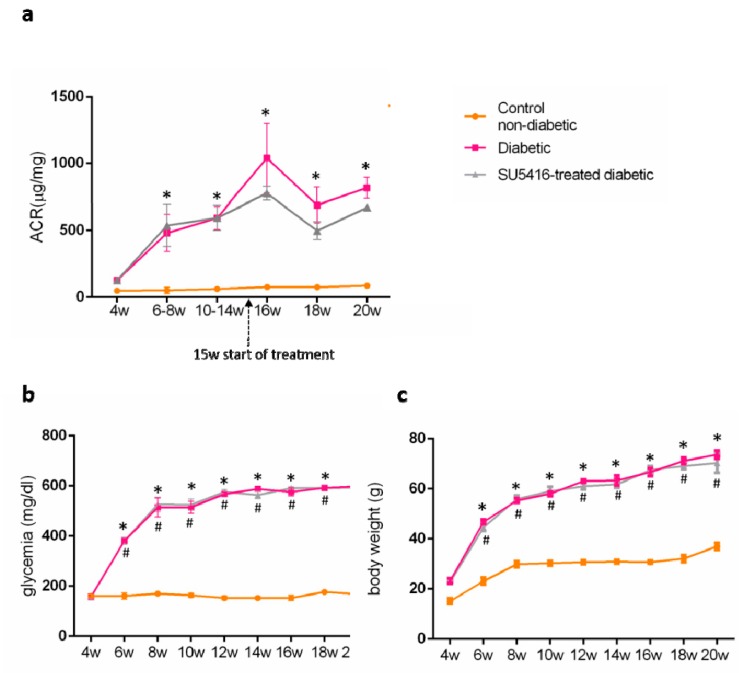
Effect of the VEGFR2 blockade on biochemical parameters in experimental diabetes. BTBR ob/ob mice (diabetic mice) were studied from the 4th to 20th weeks of age, and biochemical time-course determinations were done every 2 weeks. Occurring at 15 weeks, mice were randomly distributed into 2 groups: mice were treated with VEGFR2 kinase inhibitor SU5416 and vehicle-treated mice (diabetic group) and studied until week 20 (blood and kidney determinations were performed at the time of mice sacrifice). As a control group, non-diabetic mice (BTBR WT mice) were also studied. (**a**) Evolution of albuminuria determined as the albuminuria/creatinine ratio (ACR). Assessment over time of (**b**) glycemia and (**c**) body weight. Data as mean ± SEM of 6–8 mice per group. * *p* < 0.05 versus control non-diabetic, # *p* < 0.05 versus diabetic.

**Figure 3 jcm-09-00302-f003:**
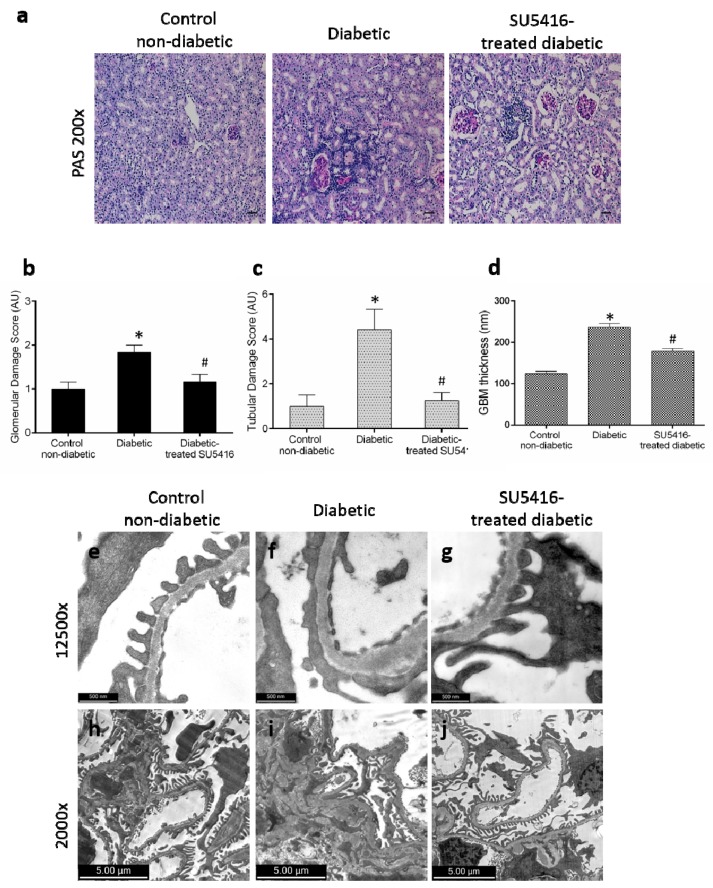
Effect of the blockade of VEGFR2 on renal lesions in a BTBR ob/ob experimental model. The BTBR ob/ob mice were treated with the VEGFR2 kinase inhibitor SU5416 (i.p, 0.1 mg/mouse, 3 times a week) (SU5416-treated diabetic group) or vehicle-treated mice (diabetic group), starting at 15 weeks and followed until 20 weeks. Renal lesions were evaluated at 20 weeks in paraffin-embedded tissue sections. (**a**) Representative images of light microscopy of kidney sections in each group stained with periodic acid–Schiff (PAS). Magnification: ×200. Bars = 20 µm. (**b**) Quantification of glomerular and (**c**) tubular damage. (**d**) The quantitative measure of glomerular basement membrane (GBM) thickness. Electron micrographs of glomeruli of representative animals for (**e**,**h**) control mice, (**f**,**i**) diabetic mice and (**g**,**j**) SU5416-treated diabetic mice (**e**–**g**) 1 µm, magnification ×12,500 and (**h**–**j**) 2 µm, magnification ×2000. Data as mean ± SEM of 6–8 mice per group. * *p* < 0.05 versus control non-diabetic, # *p* < 0.05 versus diabetic.

**Figure 4 jcm-09-00302-f004:**
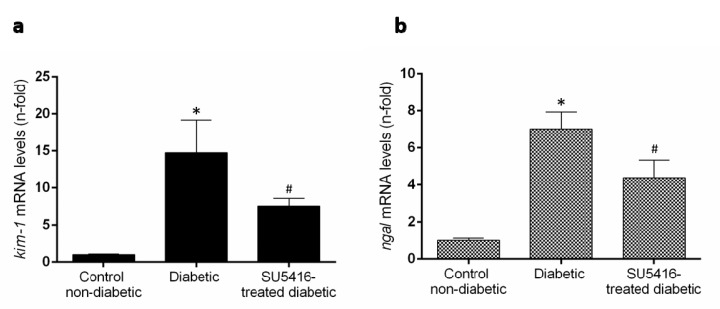
VEGFR2 blockade diminished overexpression of damage markers in murine diabetic kidneys. Gene expression levels of the biomarkers of renal damage, (**a**) kidney injury molecule-1 (*kim-1*) and (**b**) neutrophil gelatinase-associated lipocalin (*ngal*) were assessed at the end of the study at 20 weeks. Data as mean ± SEM of 6–8 mice per group. * *p* < 0.05 versus control non-diabetic, # *p* < 0.05 versus diabetic.

**Figure 5 jcm-09-00302-f005:**
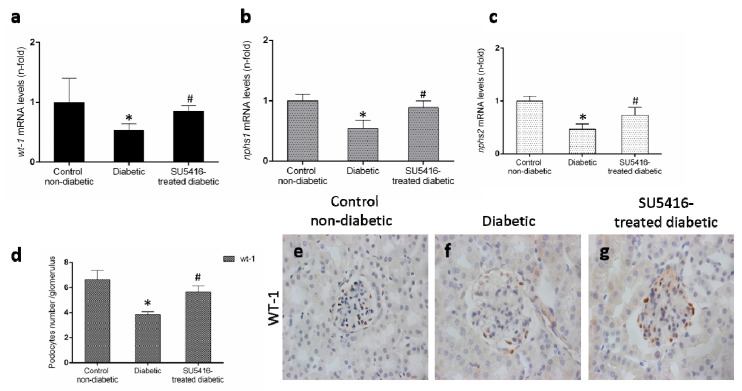
VEGFR2 kinase inhibition restores podocyte damage induced by experimental diabetes. Gene expression levels of (**a**) *wt-1*, (**b**) *nphs1*, and (**c**) *nphs2* were evaluated by real-time polymerase chain reaction. mRNA levels in each sample were normalized by cyclophilin 1. Occurring at each time point, data were normalized by the mean value of their corresponding controls. (**d**) The average number of podocytes observed of a total of 25 glomeruli per animal, and the mean ± SEM of control non-diabetic mice, diabetic mice, and SU5416-treated diabetic groups. Representative images of immunohistochemistry against podocyte marker Wilms tumor protein 1 (WT-1; brown stained nuclei) for (**e**) control non-diabetic mice, (**f**) diabetic mice and (**g**) SU5416-treated diabetic mice. Magnification ×400. Bars = 20 µm. * *p* < 0.05 versus control. Data as mean ± SEM of 6–8 mice per group. * *p* < 0.05 versus control non-diabetic, # *p* < 0.05 versus diabetic.

**Figure 6 jcm-09-00302-f006:**
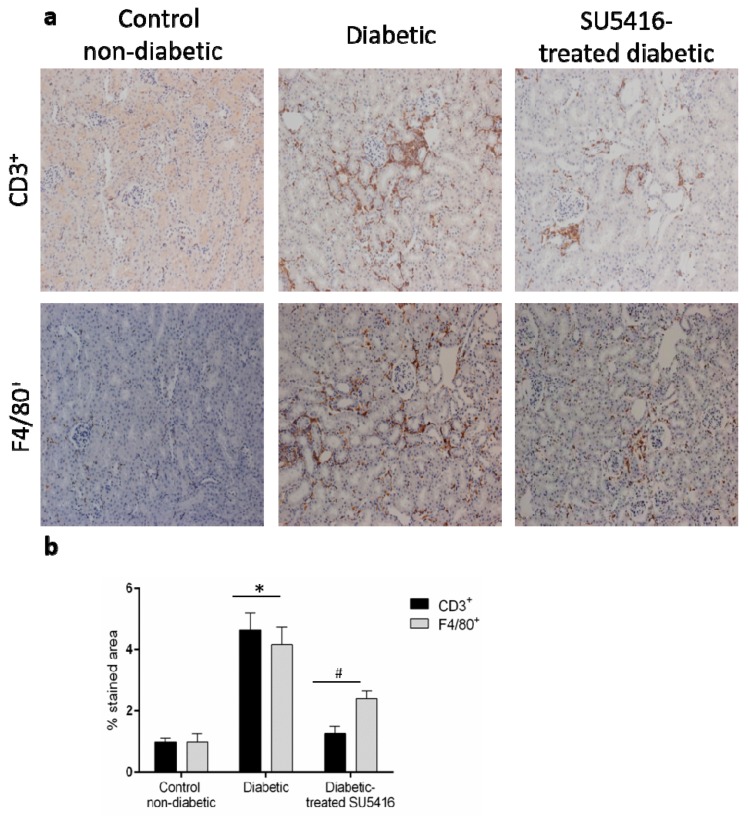
A VEGFR2 blockade inhibited renal-inflammatory responses in the BTBR ob/ob experimental model. (**a**) Quantification of F4/80+, CD3+, and CD4+ cells as described in the Methods section, as an n-fold change of positive staining versus total area, normalized by values of control mice and data of each animal and mean ± SEM. (**b**) Representative images of the immunohistochemistry for non-diabetic, diabetic and SU5416-treated diabetic animals. Magnification ×200. Bars = 20 μm. Data as mean ± SEM of 6–8 mice per group. * *p* < 0.05 versus control non-diabetic, # *p* < 0.05 versus diabetic.

**Figure 7 jcm-09-00302-f007:**
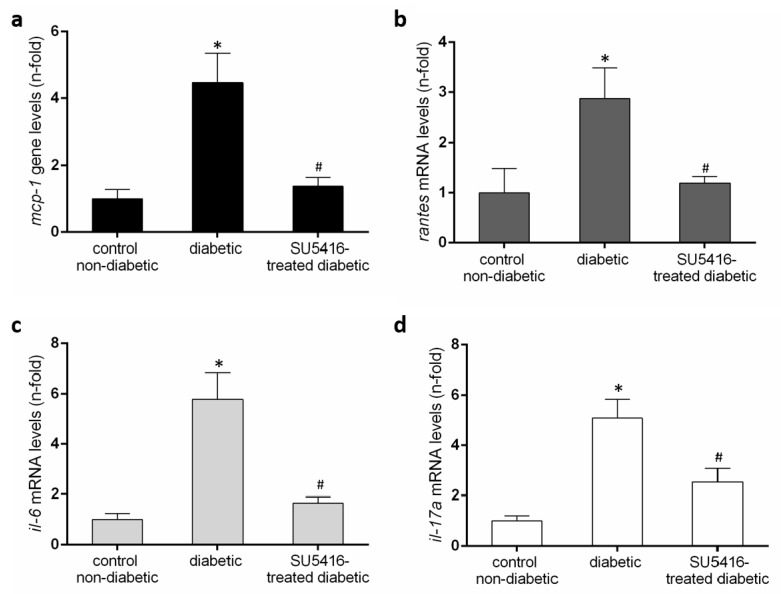
Renal chemokines mRNA expression (**a**) monocyte chemoattractant protein-1 (*mcp-1*), (**b**) regulated on activation, T cell expressed, and secreted (rantes), (**c**) *il-6* and (**d**) *il-17a* in each sample, normalized by *cyclophilin 1*. * *p* < 0.05 versus control. Data as mean ± SEM of 6–8 mice per group. * *p* < 0.05 versus control non- diabetic, # *p* < 0.05 versus diabetic.

**Figure 8 jcm-09-00302-f008:**
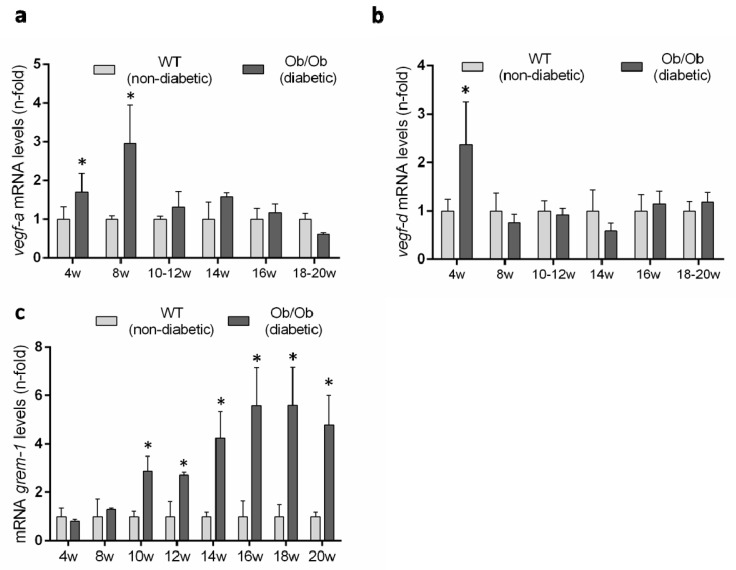
Gene expression levels of VEGFR ligands in the BTBR ob/ob experimental model. (**a**) *vegf-a*, (**b**) *vegf-d* and (**c**) *grem-1* were evaluated by a real-time polymerase chain reaction. mRNA levels in each sample were normalized by cyclophilin 1. Occurring at each time point, data were normalized by the mean value of their corresponding controls. Data as mean ± SEM of 6–8 mice per group. * *p* < 0.05 versus control non- diabetic, # *p* < 0.05 versus diabetic.

**Figure 9 jcm-09-00302-f009:**
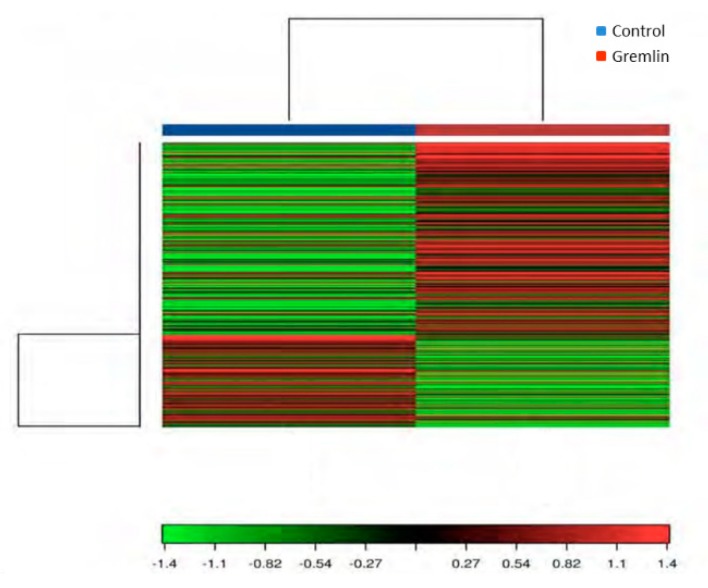
Heat map and unsupervised hierarchical clustering by sample and genes performed on the listed samples using the 500 genes that have the largest coefficient of variation rate based on FPKM (Fragments Per Kilobase of transcript per Million mapped reads) counts. Data is based on pool samples from the Control and Gremlin groups. Data were normalized using FPKM (abundance) for each gene for each sample.

**Table 1 jcm-09-00302-t001:** Primers used for quantitative polymerase chain reaction.

Gene	Forward	Reverse
*kim-1*	5-′TGTCGAGTGGAGATTCCTGGATGGT-3′	5-′GGTCTTCCTGTAGCTGTGGGCC-3′
*ngal*	5′-GCCCTGAGTGTCATGTGTCT-3′	5′-GAACTGATCGCTCCGGAAGT-3´
*wt-1*	5′-CAGCGAAAGTTTTCCCGGTC-3´	5′-TGTTGTGATGGCGGACCAAT-3′
*nphs1*	5´-AGGGTCGGAGGATCGAA-3′	5′-GGGAAGCTGGGGACTGAAGT-3′
*nphs2*	5′-CCAAAGTGCGGGTGATTGC-3′	5′-TGATGCTCCCTTGTGCTCTG-3′
*mcp-1*	5′- AGCTCTCTCTTCCTCCACCA-3′	5′ -GGCGTTAACTGCATCTGGCT-3
*rantes*	5′-AGAGGACTCTGAGACAGCACA-3′	5′- CGAGCCATATGGTGAGGCAG-3′
*il-6*	5′CCCCAATTTCCAATGCTCTCC-3´	5′-CGCACTAGGTTTGCCGAGTA-3′
*il-17a*	5′-TCTCCACCGCAATGAAGACC-3′	5′-GACCAGGATCTCTTGCTGGA-3′
*grem-1*	5′-CCCGGGGAGGAGGTGCTGGAGT-3′	5′-CCGGATGTGCCTGGGGATGTAGAA-3′
*vegf-a*	5′-TATTCAGCGGACTCACCAGC-3′	5′-AACCAACCTCCTCAAACCGT-3′
*vegf-d*	5′-TGTGACGCAGTGAGGGAGT-3′	5′- AAGCAGTTGTTTCCAGGCAG-3′
*cyc*	5′-GGCAATGCTGGACCAAACACAA-3′	5′-GTAAAATGCCCGCAAGTCAAAAG-3′

Abbreviations: *il*, interleukin; *kim-1*, kidney injury molecule-1; *ngal*, neutrophil gelatinase-associated lipocalin; *wt-1*, Wilms tumor protein-1; *nphs1*, nephrin; *nphs2*, podocin; *mcp-1*, monocyte chemoattractant protein-1; *rantes*, regulated on activation, T cell expressed, and secreted; *grem-1*, gremlin-1; *vegf*, vascular endothelial growth factor; *cyc*, cyclophilin.

**Table 2 jcm-09-00302-t002:** Genes regulated by Gremlin in the kidney. Renal transcriptomic changes were evaluated by RNA sequencing comparing kidneys of control and Gremlin-treated mice. Samples from 5 samples per group were pooled and analyzed as described in Methods. The Table shows genes with >2.5-fold upregulation in Gremlin vs. control mice.

Gene	Control FPKM	Gremlin FPKM	Log2 Fold Change	*q* Value
***Havcr1***	1.83792	65.521	5.15581	0.0636633
*AC131780.3*	1.40074	49.7095	5.14926	0.0636633
*Gm10800*	10.8964	212.665	4.28666	0.0636633
*Serpine1*	0.400392	5.56466	3.79681	0.0636633
***Lcn2***	7.64414	98.5245	3.68806	0.0636633
*Aldh1a3*	1.16126	12.0796	3.37881	0.0636633
*Serpina10*	2.62027	22.3256	3.09091	0.0636633
*Col1a1*	1.29665	10.9816	3.08223	0.0636633
*Ugt2b37*	1.12231	7.54508	2.74907	0.0636633
*Col3a1*	4.35433	27.7459	2.67175	0.0636633
*Mki67*	0.606635	2.70504	2.15675	0.0636633
*C3*	2.35343	10.4619	2.1523	0.0636633
*Cdk1*	1.80221	7.87254	2.12706	0.0636633
*Fgg*	6.5632	27.6495	2.07478	0.0636633
*Igfbp2*	3.86836	14.4286	1.89914	0.0636633
*Serpina1a*	2.53818	9.26732	1.86836	0.0636633
*Tnc*	0.777548	2.79985	1.84834	0.0636633
*Spp1*	2049.39	5757.69	1.4903	0.0636633
*Vcam1*	2.53223	7.00542	1.46806	0.0636633
*Mgp*	134.793	326.969	1.27841	0.0636633
*Cfi*	6.66434	16.1511	1.2771	0.0636633
*Lyz2*	27.1578	60.4045	1.15329	0.0636633
*Fabp4*	377.135	161.948	−1.21955	0.0636633
*Car3*	78.7394	11.605	−2.76233	0.0636633
*Thrsp*	35.0519	3.21214	−3.44789	0.0636633
*Retn*	22.4473	1.61617	−3.79589	0.0636633
*Adipoq*	16.8479	0.961797	−4.13069	0.0636633
*Cfd*	148.864	7.47942	−4.31493	0.0636633

**Table 3 jcm-09-00302-t003:** Gene Ontology (GO) analysis was done with the most upregulated genes (see Table 2) in Gremlin-treated mice. This Table shows the most deregulated Cellular Component (CC), Biological Process (BP), Molecular Function (MF) and KEGG pathways. Control and Gremlin columns are group average FPKM values. FPKM is a unit of measuring gene expression. Fold change is the log2 fold change of the FPKM between the Control and Gremlin groups. *q*-values shown are *p*-values that have been adjusted using the Benjamini–Hochberg False Discovery Rate (FDR) approach to correct for multiple testing.

**Category**	**Term**	**Count**	***p*-Value**
GOTERM_BP_DIRECT	Response to ethanol	5	1.60E−05
GOTERM_BP_DIRECT	Response to mechanical stimulus	4	7.70E−05
GOTERM_BP_DIRECT	Wound healing	4	2.50E−04
GOTERM_BP_DIRECT	Cellular response to tumor necrosis factor	4	4.00E−04
GOTERM_BP_DIRECT	Aging	4	1.50E−03
GOTERM_BP_DIRECT	Regulation of triglyceride biosynthetic process	2	4.00E−03
GOTERM_BP_DIRECT	Response to nutrient	3	4.20E−03
GOTERM_BP_DIRECT	Cell-matrix adhesion	3	4.80E−03
GOTERM_BP_DIRECT	Blood coagulation	3	5.50E−03
GOTERM_BP_DIRECT	Osteoblast differentiation	3	9.80E−03
GOTERM_BP_DIRECT	Response to drug	4	9.90E−03
GOTERM_BP_DIRECT	Immune system process	4	1.40E−02
GOTERM_BP_DIRECT	Positive regulation of gene expression	4	1.50E−02
GOTERM_BP_DIRECT	Innate immune response	4	1.50E−02
GOTERM_BP_DIRECT	Negative regulation of smooth muscle cell migration	2	2.10E−02
GOTERM_BP_DIRECT	Protein heterotrimerization	2	2.50E−02
GOTERM_BP_DIRECT	Negative regulation of extrinsic apoptotic signaling pathway via death domain receptors	2	3.30E−02
GOTERM_BP_DIRECT	Positive regulation of interleukin-8 production	2	3.30E−02
GOTERM_BP_DIRECT	Negative regulation of endothelial cell apoptotic process	2	3.70E−02
GOTERM_BP_DIRECT	Brown fat cell differentiation	2	4.40E−02
**Category**	**Term**	**Count**	***p*-Value**
GOTERM_CC_DIRECT	Extracellular space	17	5.20E-13
GOTERM_CC_DIRECT	Extracellular region	16	1.00E-10
GOTERM_CC_DIRECT	Extracellular exosome	17	3.30E−09
GOTERM_CC_DIRECT	Extracellular matrix	5	4.10E−04
GOTERM_CC_DIRECT	Collagen trimer	3	4.60E−03
GOTERM_CC_DIRECT	Endoplasmic reticulum	6	2.00E−02
GOTERM_CC_DIRECT	Cell surface	4	4.00E−02
**Category**	**Term**	**Count**	***p*-Value**
GOTERM_MF_DIRECT	Extracellular matrix structural constituent	3	1.40E−03
GOTERM_MF_DIRECT	Protease binding	3	1.10E−02
GOTERM_MF_DIRECT	Serine-type endopeptidase inhibitor activity	3	1.20E−02
GOTERM_MF_DIRECT	Platelet-derived growth factor binding	2	1.60E−02
GOTERM_MF_DIRECT	Protein homodimerization activity	5	2.20E−02
GOTERM_MF_DIRECT	Endopeptidase inhibitor activity	2	4.20E−02
**Category**	**Term**	**Count**	***p*-Value**
KEGG_PATHWAY	Complement and coagulation cascades	5	1.50E−05
KEGG_PATHWAY	ECM-receptor interaction	4	7.30E−04
KEGG_PATHWAY	Staphylococcus aureus infection	3	4.70E−03
KEGG_PATHWAY	Focal adhesion	4	8.30E−03
KEGG_PATHWAY	Platelet activation	3	3.00E−02
KEGG_PATHWAY	PI3K-Akt signaling pathway	4	3.40E−02

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
