# Peer review of "VEGFR2 Blockade Improves Renal Damage in an Experimental Model of Type 2 Diabetic Nephropathy"

_jcm, 2020, doi:10.3390/jcm9020302_

Round 1

Reviewer 1 Report

The article by Lavoz C et al describes the therapeutic potential of blocking VEGFR2 signaling pathway in a rodent model of diabetic nephropathy on BTBR background. The authors report promising results where VEGFR2 kinase inhibitor, SU5416 reversed podocyte damage, albuminuria and inflammation in established diabetic nephropathy.

The authors show that VEGFA levels increased during the early phase of nephropathy while they did not change after 10 weeks. The authors should discuss how blockade of VEGFR2 signaling starting from 15 weeks resulted in beneficial effects. Would it make any difference if the treatment started at an early stage, which might prove the causal role of VEGFR2 signaling in kidney damage. 

2. Are there inhibitors available to block GREMLIN signaling in the kidneys. Would that open up more treatment options for nephropathy. Please discuss this. 

3. The authors may consider splitting the gene expression figures to show individual gene expression changes between the groups rather than pooling all the genes together in Fig. 4a, 5a and 6c. 

4. Correct the sentence in line 72 and 521. 

Reviewer 2 Report

- The introduction section should be rewritten. Please combine the sections concerning the VEGF ligands in one section.

Page 3, line 130: The number of animals in each group should be included. “1) non diabetic (named control), 2) vehicle-treated diabetic (named untreated-diabetic), …”

Page 5, line 228: The sentence “To investigate whether VEGFR2 signaling was involved in the progression of renal diabetic damage, BTBR ob/ob mice were treated with the VEGFR2 kinase inhibitor SU5416, starting at 15 weeks. At this time point, in which renal lesions resemble established human DN, VEGFR2 kinase inhibition was carried out as a therapeutic approach, and then mice were followed until the 20th week.” does not belong to the results section, but to the methods section.

Page 6, line 233: The sentence “Previous studies have described activation of VEGFR2 pathway in experimental diabetes [4,5], 233 but there is no data in the BTBR ob/ob model.” should not be included in the results section, but in the discussion section.

Page 7: Figure 2 shows the ACRs in the different study groups. It is not clear why in the X-axis, weeks 6-8 and 10-14 weeks were combined. I would like to see the separate values at week 6-8-10-12-14.

Page 7, line 307: The sentence “Changes in renal structure in response to VEGFR2 blockade were studied at 20 weeks. The 307 tissue damage score was calculated as described [31], including the degree of glomerular sclerosis, 308 increased mesangial matrix, hyalinosis, tubular casts, acute tubular damage, and tubular atrophy, as 309 well as the presence of interstitial inflammatory cells.” does not belong to the results section, but to the methods section.

Page 8: Figure 3b and 3c should also include a box containing the values in non-diabetic mice.

- The paper contains several grammatical errors and should be rewritten by a native English speaker.

Round 2

Reviewer 2 Report

Paper can be accepted in its present form.